# The Impact of Industry 4.0 on the Steel Sector: Paving the Way for a Disruptive Digital and Ecological Transformation

**Laura Tolettini** [1,2,*] and **Eleonora Di Maria** [2]

1    Feralpi Siderurgica SpA, 25017 Lonato Del Garda, Italy
2    Department of Economics and Management, Università Degli Studi di Padova, 35121 Padova, Italy
*    Correspondence: laura.tolettini@studenti.unipd.it

**Abstract:** Since the creation of a common term to indicate a set of incremental and disruptive digital technologies, Industry 4.0 has challenged European manufacturers to find a way to concretely exploit these innovations in their own business strategy. During this journey, Industry 4.0 has recently highlighted some evidence about its efficacy in enabling strategic goals on the three dimensions (economical, environmental, social) of sustainable development, which is a key element for the European Union's goal to make manufacturers become carbon neutral until 2030. Industry 4.0 and sustainability are together affecting manufacturers' business models, forcing managers to take chances and face challenges within their organization and in their supply-chain. As an energy-intensive sector, steel industries will be intensively affected by sustainability paradigms. With 19 qualitative interviews in the organization and supply chain of an internationalized steel producer, Feralpi Group, we provide evidence that, beyond the use of main strategic technologies (Internet of Things and Big Data analysis), the implementation of a sustainability strategy is also possible through the creation of new partnerships beyond the own supply chain. The combination of Industry 4.0 technologies and sustainability strategies, especially concerning the environment through Circular Economy practices, pushes steel industries to revise their business models, paving the way for unexpected collaborations, where suppliers, customers, and even more diverse stakeholders such as competitors could bring benefits to the company sustainable economic growth and durability.

**Keywords:** Industry 4.0; sustainability; circular economy; business model innovation; network collaboration





## 1. Introduction

It is unfortunately well known how COVID-19 ignited one of the most terrible economic crises since World War II, affecting more than 50 million people and with more than 1.25 million victims around the globe [1]. In 2020, there was a contraction of $-3.4\%$ in the global economy, with uncertain projections of growth due to virus variants spreading and incomplete vaccine coverage globally [2]. All around the world, governments started economic initiatives in order to boost economic growth and to provide a strategic direction in these conditions of uncertainty. Many of these actions were intended to ignite technological innovation, with the goal to incentivize more sustainable business models. In the United States, President Joe Biden promised to invest $300 billion in breakthrough technologies (lightweight materials, 5G, artificial intelligence, electric vehicles) [3] in order to fulfill the goals of the rejoined Paris Agreement, signed in December 2015 by 190 parties, to limit global warming to 2 °C [4].

In April 2020, as a result of the negotiation for a multiannual financial framework (€1800 billion in 2020–2027), the European Commission approved a recovery instrument called "Next Generation EU" (European Union)—NGEU, injecting €500 billion in grants and €250 billion in loans for member states [5]. The goal of this strategic plan is to push EU companies toward more sustainable business models with the support of digital and

Industry 4.0 technologies. Each member state has to dedicate at least 37% of the national expense to the reduction of climate impact and 20% to the empowerment of the digital transition [6]. Prime Minister Draghi has presented a total investment plan of €248 billion €, with 40% dedicated to the ecologic transition and 27% to digitization and innovation [7]. The digital and green economy transitions are the two most important missions of the Italian recovery plan. The digital transformation, especially based on high connectivity technologies, should foster the competitive advantage of Italian industries at an international level. The ecologic transition should have energy efficiency, the use of renewable energy, and the reuse of regenerated and recycled materials as core drivers [8]. Germany has also focused on investing 90% of its resources in green and digital technologies, although with a smaller amount compared to Italy (€11.5 billion and €14 billion, respectively) [9,10].

The EU Green Deal and Climate Action Plan contains very challenging goals until 2030 for the EU manufacturing industries, in order to become a climate-neutral economy such as [11]:

−55% reduction in greenhouse gas emissions (compared to 1990);
+32% renewable energy;
+32.5% improvement in energy efficiency

Technology has incrementally supported EU manufacturers to control industrial emissions and to reduce environmental impact, by employing the so-called Best Available Techniques (BAT) [12]. In particular, in the last few years, a set of key technologies centered on the use of embedded Cyber-Physical-Systems (CPS) under the term Industry 4.0 can connect machines, products, and people and exchange real-time information through high-speed wireless networks [13]. CPS connect physical and software systems to integrate the cyber and physical world [14].

Recent research has developed on the importance of Industry 4.0 as an enabler of sustainable development, intended as the human strategic commitment to manage and preserve ecosystem resources in order to fulfill current and future generations' needs [15]. Sustainability can transform companies' business models, defined as the way companies make value from their economic activities, overcoming challenges by fostering the convergence of multiple stakeholders' interests and tensions [16]. A business model is the way a company defines its competitive strategy, through its products and services offered to the market, in a unique value proposition that is differentiated from competitors [17].

Sustainable business models consider the needs of diverse stakeholders and measure performance based on three levels, economic, social, and environmental, the so-called triple bottom line [18]. Sustainable economic activities encourage an optimized use of resources, such as raw materials and energy, and integrate social and environmental scopes, rather than only prioritizing consumerist growth; ecological, social, and economic goals are fully integrated [19]. Specifically, at the environmental level, Circular Economy (CE) represents an important sustainability paradigm, where environmental innovation can enable durable competitive advantage and successful economic development from a long-term perspective [20], and support an important integration among the social, environmental, and macroeconomic dimensions [21].

CE could be considered a part of the sustainability strategy. CE is especially focused on the economic and environmental dimensions, which can benefit from each other, while the overall concept of sustainability looks at the interest alignment of all stakeholders on the whole triple bottom line [22]. CE is a sustainability archetype affecting the technological sphere, pushing companies to find new technical routes to reuse and recycle apparently exhausted materials [23]. More specifically, CE implies a "closed-loop" ecosystem, where waste is not generated or significantly minimized, (in the case of manufacturing industries, we can speak of "zero waste" production strategies) [24,25], and where customers' needs are completely aligned with suppliers' circular activities [26,27].

It has been often investigated and confirmed how the power of digitization and interconnection offered by Industry 4.0 technologies can shape sustainable manufacturing [28,29]. However, this has been mostly carried out on a theoretical basis and literature

review level, leaving space for further research to dig deeper in relation to these technologies with sustainability strategies in concrete manufacturers' examples [30]. The focus can be particularly set on all three levels of sustainable business models, not only on the environmental aspect [31] but also on the social and economic dimension [32,33], while highlighting the concrete benefits and challenges of those technologies. This is the literature gap of our research, with the main novelty of investigating in a sector—steel industries—which was almost unexplored from this point of view. Industry 4.0 and digitization seem to become more and more a strategic enabler of sustainability performance [34].

Steel industries are highly impacted by the EU green economy and sustainability goals. Steel industries represent the backbone of the European economy and they are the most advanced in the energy-intensive sector in terms of the opportunities given by environmental sustainability, with the use of breakthrough technologies such as hydrogen- and electricity-based metallurgy and carbon capture and storage [35]. However, this sector remains almost unexplored regarding how Industry 4.0 can enable and shape the strategy of these industries, not only concerning the optimization and reuse of resources, under a CE umbrella but also the further exploration of their sustainability strategy as a whole [36,37]. This analysis represents an important point in the current research on Industry 4.0 and sustainability since steel industries will be powerfully affected by the digital and decarbonization transformation [38]. Analyzing the case of a leading internationalized company in the steel branch, we take an innovative focus on the managers' perspective, who are the decisive protagonists for Industry 4.0 and sustainability practices in manufacturing companies and who are the ultimate decision-makers in the fulfillment of digitization processes in steel organizations [39]. In this sense, our explorative research question is twofold:

RQ1: What is the prioritized sustainability strategy in the steel sector? Which are the benefits and challenges brought by Industry 4.0 technologies in order to pursue this strategy in the steel sector?

RQ2: Are Industry 4.0 and sustainable development causing radical changes in the business model of such a traditional and long-established energy-intensive sector? If yes, how?

In the following sections, we will go deeper into those themes: in Section 2, we will present the literature review; in Section 3, methodology; in Section 4, the main findings; in Section 5, the conclusion and insight into future research.

## 2. Literature Review

### 2.1. Industry 4.0 and the Path towards Sustainability

Industry 4.0 is a German term (Industrie 4.0)—launched at the Hannover Fair in 2011—in order to characterize the ongoing fourth industrial revolution affecting industries with the use of the Internet of Things, Data, and Services [40]. Industry 4.0 collects a set of enabling technologies leading to the efficient integration and interconnection of internal production processes and to a tighter collaboration among different actors in the external ecosystems of manufacturing companies [41,42]. Integrated connectivity is the basic design principle of Industry 4.0, making internal production ecosystems collect data in real time and facilitate decision-making [43].

Compared to the previous industrial innovation streams, the connectivity enabled by Industry 4.0 fosters information availability and transparency, knowledge sharing [44], and a decentralized and automated integrated decision-making process [45]. The literature identifies nine enabling technologies; most of them have already been developed through the years [46,47]:

1.  Robotics: with a great range of capabilities, robots can fulfill tasks autonomously and even collaborate with humans in the same working environment, facilitating strenuous and dangerous activities;

2.  Simulation: time for machine set-up and product as well as process configuration can be reduced through virtual simulations, which can replicate real conditions very precisely;
3.  Vertical and horizontal systems integration: engineering and automation make production phases and systems more integrated; digital platforms enable the real-time information exchange in the supply chain;
4.  Industrial Internet of Things (IoT): sophisticated sensors and embedded cyber-physical systems make machines and products interact with each other and human beings by employing wireless communication;
5.  Cybersecurity: articulated and encrypted protocols, control, and identification procedures should protect real-time connected systems from cyber-crime attacks, which can endanger production and business continuity;
6.  Cloud: cloud-based software enables high-speed analytics and data-driven manufacturing systems, decreasing maintenance costs and enhancing Information Technology architecture security;
7.  Additive manufacturing: 3D printing leads to customized product design, supporting manufacturing decentralization and stock reduction and increases the lightness of material weight;
8.  Augmented reality: virtual reality is connected to the concrete world through human–machine interfaces (like glasses and tablets), creating opportunities for better maintenance and training directly on the plant.
9.  Big Data Analytics: from different heterogeneous sources large amounts of data can be collected and interpreted in order to gain internal efficiency and durable competitive advantage.

These technologies are pushing manufacturers to reshape their business models, specifically, the way they conduct their business and create value with their key business activities [48]. They pressure them to integrate their physical product portfolio with more services and to satisfy customers' needs more individually by offering new value propositions [49]. Industry 4.0 affects manufacturers' business paradigms at various levels: from a direct technological impact with organizational consequences to a pervasive evolution, with a radical transformation of the business model to adapt to the new complex digital ecosystem [50,51]. Since existing business models are put into discussion, manufacturers have to face substantial challenges, from a technical and organizational perspective, such as business future viability, production fit (high technical and financial implementation efforts), and employee acceptance (employee fear and concern to be substituted by autonomous technologies) [52–54]. However, Industry 4.0 leads manufacturers to undoubted significant competitive advantages: a 10–40% reduction in maintenance costs, a 20–50% reduction in time to market [55], an 85% increase in forecast accuracy, and a 3–5% increase in productivity [56]. Table 1 shows a more detailed list of the opportunities and challenges of Industry 4.0 technologies [57].

Recent research has underlined how Industry 4.0 can be a facilitator of sustainability strategies, especially in the generation of CE practices from the perspective of a supply-chain integrated collaboration [58,59]. Industry 4.0 technologies cause benefits affecting all the triple-bottom-line dimensions: economic (increased productivity); ecological (increased resource efficiency); social (increased job opportunities) [60], giving impulse to manufacturers for sustainable development [61]. Sustainability reshapes the financial and organizational architecture of firms and can characterize the firm in new business archetypes [23]: technological (e.g., use of innovative technology for low carbon and energy-efficient production, use of renewable resources in the process to ignite the best practice of reuse, recycle, remanufacture); social (e.g., stewardship for sustainable growth, like ethical business and consumer education); organizational (e.g., multi-stakeholder collaboration). Sustainable business models are driven by the triple-bottom-line approach (social, economic, environmental) to assess performance, considering all stakeholders' needs including those of the surrounding environment [18]. Sustainable business models

are attained through the consideration of the interests of all stakeholders in the value chain. An important aspect to consider is the identification and tackling of conflicting values when the interests of some stakeholders could damage those of other parties [62].

**Table 1.** Main opportunities and challenges of Industry-4.0-related technologies.

| Main Chances and Challenges of Industry 4.0-Related Technologies | | |
|---|---|---|
| | **Chances** | **Challenges** |
| IoT | Production integration and control | Integration with existing technology |
| Advances in Automation (Robots and Vertical Integration) | Enhanced safety, standardization, speed | Cooperation with human operators |
| Data Platforms (horizontal integration) | Data integration in the supply chain | Homogeneous digital level of maturity of own business partners |
| Cybersecurity | Business continuity | Decision on the level of systems security |
| Additive Manufacturing | Customized production | Still higher production costs than the standards processes |
| Simulation | Spare of prototypes and resources | Setting the right parameters |
| The Cloud | Integration of all technological devices | Data privacy control |
| Augmented Reality | Simulation of dangerous activities | Still implementation costs |
| Big Data | Live information availability | Data interpretation and reliability |

The economic and environmental dimensions of sustainability have become even more specifically intertwined thanks to the recent development of sustainable financing. In May 2021, the European Banking Authority (EBA) published a report that officially stated the importance of sustainability criteria (Environmental, Social, and Governance—ESG) at the base of financing. ESG criteria are a new important factor for banks to evaluate a company's industrial plans and the riskiness of their activities. Risks drivers are valued under Environmental (e.g., the production level of gas emissions and protection of biodiversity), Social (e.g., personal data security, workplace health and safety, quality, and innovation in customers' relations), and Governance (Board of Directors independence, shareholders rights, integrity in corporate conduct) categories [63]. Another important driver for sustainable investments is the European Taxonomy Regulation, which came into force on 12th July 2020. It classifies and assesses company investments based on six main environmental objectives [64]:

1. Climate change mitigation;
2. Climate change adaptation;
3. Sustainable use and protection of water resources;
4. Circular Economy transition;
5. Pollution prevention;
6. Protection of biodiversity ecosystems.

The main benefits of Industry 4.0 in a sustainability strategy are found to be more specifically connected to the concept of Circular Economy (CE). In contrast to the traditional linear economy, CE is based on three fundamental concepts: design out waste (materials

are designed to be used again with the lowest energy consumption and highest quality retention); diverse systems based on renewable resources (resilient systems adapting and evolving in uncertain conditions); and eco-effectiveness (material flow is cradle-to-cradle, where resources are reusable and accumulate intelligence over time) [65]. Moreover, CE is a powerful ally for $CO_2$ emissions reduction and climate change mitigation [66]. Several quantitative studies and literature reviews have analyzed the connection between different Industry 4.0 technologies and Circular Economy effects. IoT and Big Data especially support the optimization of resource usage and energy consumption and emission and waste reduction along with production process optimization and enhanced product quality [67–70]. The Cloud is mainly applied in order to maximize production flexibility and customization [71].

Other research has focused on the combination of Industry 4.0 and Circular Economy in order to fulfill Sustainable Development Goals (SDGs), the 17 goals ratified by the United Nations focused on the main objectives of people, profit, and planet in order to develop a fairer more sustainable and carbon neutral world [72]. The benefits of Industry 4.0 technologies regarding CE (IoT, Big Data, Cloud) can be linked to SDG 7 (affordable and renewable energy), SDG 8 (resource efficiency through new circular business paradigms), SDG 12 (minimize waste production through redesign, reuse, and recycling), and SDG 13 (climate mitigation) [73]. However, there is still space to explore the real connections between Industry 4.0 technologies and sustainability strategy in manufacturing industries [74], looking at the impact of digitization on the triple-bottom-line dimension (environmental, economic, social) [75].

### 2.2. Digitization and Sustainable Business Models in the Steel Sector

There is space for deeper research on how Industry 4.0 and digitization can affect the business model of manufacturing industries and their supply chains [76], especially for energy-intensive manufacturers [37], and how these technologies can enable sustainability strategies involving the entire triple-bottom-line approach, focusing particularly on the environmental dimension [77]. In particular, the steel sector can offer an interesting research landscape to analyze Industry 4.0 in terms of its opportunities and challenges in the context of sustainability for energy-intensive industries. European and international regulations concerning low carbon emissions are putting pressure on steel industries. These industries are increasingly employing intelligent digital systems in order to fulfill not only strictly environmental constraints but also to align with the interests of more diverse internal and external stakeholders such as employees and local communities [78]. Industry 4.0 and digitization have often been viewed as powerful enablers of decarbonization and environmental sustainability strategy in the steel sector, enhancing innovation and flexibility [79]. Due to their supply-chain structure and product lifecycle, different projects with the objective of zero waste impact were being developed even before the boom of Industry 4.0 initiatives [80], taking advantage of the economic and environmental opportunities offered by conscious sustainability strategies.

Steel is a continuously recyclable material. The recyclability of steel is of strategic importance for our daily lives, not only from a circularity point of view but also for climate change mitigation. If we increase current steel recyclability, we could avoid 500 million tons of additional primary steel production by 2050, sparing 1 billion tons of $CO_2$ emissions per year [66]. For Electric Arc Furnace (EAF) steel manufacturers, scrap is the fundamental primary raw material for production, enabling a more eco-friendly manufacturing process [36]. With almost 2 billion tons of crude steel production worldwide in 2019, steel is a recyclable material present in all domains of human life (52% building and infrastructure, 16% mechanical equipment, 12% automotive, 10% metal products, 5% transportation, 3% electrical equipment, 2% domestic appliances) [81]. Steel companies are challenged in their business, technology, process, and organization by the decarbonization strategy. Representing 8% of global carbon dioxide emissions; carbon-friendly steel products will require more engagement in sustainable trajectories and in digitally driven disruptive

technologies [82]. Digitization stays a top priority for steel producers to achieve environmental sustainability, across the complete value chain (from procurement, with automated contract management; to production, with predictive maintenance; to sales, with facilitated customer experience) [83]. Sustainability could present steel industries with an opportunity to guarantee a durable business model, which is now faced cyclically by other important threats such as the dramatic price fluctuation of transportation and raw materials [84]. A more efficient production cycle can support low-carbon manufacturing for a material like steel, whose product lifecycle is much longer than other components. For construction, the long durability of steel facilitates $CO_2$ emission reduction [85]. Steel supports sustainable construction models, which can be enabled also by the use of new green materials such as fine volcanic ashes [86] and nano-silica-modified concrete [87]. A fundamental aspect of research in the steel sector is to discover how sustainability strategies ignite Industry 4.0 applications and vice versa [34]. In the following section, we will present the methodology for our research development.

## 3. Methodology

Since the steel sector is still a broad field for our research questions, we decided to use qualitative research, particularly, the case of a specific company, in order to gain new insights about Industry 4.0 benefits and the impact on steel manufacturer business models and practices in the context of sustainable growth [88]. The selection of the case of the Feralpi Group is due to many significant reasons: its internationalization, giving insights on different cultural perspectives [89]; its sustainability engagement on the three dimensions (economic, environmental, and social) is rooted in its origins ("Produce and growth in respect of human beings and the environment", the vision of the founder since 1968); and it is officially externally recognized, with different awards and acknowledgments (Italian Stock Exchange Oscar 2022 for the voluntary non-financial disclosure and the Financial Times Climate Leaders recognition for technological innovation and a long history of sustainability) [90,91]. It also has an innovative sustainable financing strategy, with the first CE-linked Interest Rate swap for €40 million investments linked to CE objectives [92]; its 5-year-plan climate strategy, with €100 million designated to renewable energy production [93]. For the listed reasons, the Feralpi Group represents a significant case study to be explored with qualitative analysis [94].

Feralpi Group is an internationalized leader in steel products for the construction industry and it was the first among competitors to voluntarily publish sustainability reports since 2004. It is a family company founded in 1968 in Lonato del Garda, and since 1992, it has been the first steel producer of its province to have a strategic production site abroad, more precisely, Feralpi Stahl in Riesa, Eastern Germany. It is present in two of the strategic EU nations for steel production, Italy and Germany. The Group currently employs more than 1800 employees in six countries, with a steel production of 2.5 million tons, a turnover of €1.2 billion (63% realized internationally), 2.5 million tons of steel products, and net assets of €521 million. In 2020, the Feralpi Group invested €56 million in strategic initiatives, for 20 innovative Research and Development (R&D) projects. In its next industrial business plan (2021–2026), the Feralpi Group intends to invest €300 million in innovative technologies to foster the ecological and digital transformation of its international companies, with a particular focus on the German subsidiaries [95].

We employed exploratory semi-structured interviews to give interviewees a certain degree of freedom to underline possible additional aspects of the topic [96]. External companies were selected based on their level of maturity and interest in Industry 4.0 and sustainability initiatives. We pursued nineteen qualitative interviews, during two months (from 21 March 2021 to 31 May 2021). Interviews were possible both in vis-a-vis and video call modalities, due to COVID-19 restrictions, and lasted one hour on average. We interviewed thirteen directors and managers within the Feralpi Group, with three diverse perspectives to get further insight due to ethnographic diversity [97,98]: Group functions, Italian parent company leaders, and German subsidiary representatives. Interviewees were

key figures of strategic departments dealing with company, sustainability, and Industry 4.0 strategies: the Board, the CSR (Corporate Social Responsibility) department, the Italian and German environmental department, the Group Information Technology office, the Group, the Italian and German technical, production and steel department, the Group financial and administration office, and the R&D (Research & Development) Italian department. Moreover, we added two additional interviews with two business partners of the Feralpi Group, concerning Circular Economy models. They could create a new business model starting from the disposal of materials from both the product cycle inside Feralpi (the black slag resulting from the steel melting process) and outside Feralpi (polymers created from urban waste). The Feralpi Group is a shareholder of the business partner reusing the black slag. Finally, we interviewed three suppliers and one customer of the Feralpi Group, who were engaged in sustainability and Industry 4.0 strategies, to counter-check the answers given by Feralpi managers and add perspective to the topic from a more complete value-chain point of view [99,100]. In Table 2, we summarize the list of interviews, underlining their field of action and functional focus, connected to the main themes of the research: Industry 4.0, Sustainability Strategy, Circular Economy, Digitization, and Innovation.

**Table 2.** List of all interviewees and roles.

|  | Company | Department | Functional Focus |
|---|---|---|---|
| A | Italian Business Partner (black slag regeneration) | f Production | CE, Innovation |
| B | Italian Customer (Drawing Mill) | Board | Innovation, Sustainability |
| C | Italian Supplier (metal carpentry) | Board | Innovation, Sustainability |
| D | Italian Supplier (Digital Service) | Board | Digitization, Industry 4.0 |
| E | Italian Business Partner (Polymers) | Business Development Department | Innovation, CE, Digitization |
| F | German Supplier (Disposal Services) | Board | CE, Digitization |
| G | Feralpi Group | Board | Innovation, Sustainability |
| H | Feralpi Group | Group Technical department | Innovation, Industry 4.0, CE |
| I | Feralpi Group | CSR | Sustainability, CE |
| J | Feralpi Group | Information Technology | Digitization, Industry 4.0 |
| K | Feralpi Group | Board | Innovation, Sustainability, Digitization |
| L | Feralpi Group | Finance and Administration | Sustainability, CE |
| M | Feralpi Italy | Environmental department | CE, Sustainability |
| N | Feralpi Italy | Technical department | Sustainability, CE, Industry 4.0 |
| O | Feralpi Italy | Steel plant | CE, Sustainability, Industry 4.0 |

**Table 2.** *Cont.*

| | Company | Department | Functional Focus |
|---|---|---|---|
| P | Feralpi Italy | R&D | Industry 4.0, Innovation |
| Q | Feralpi Germany | Environmental department | CE, Sustainability |
| R | Feralpi Germany | Board | Sustainability, Innovation |
| S | Feralpi Germany | Technical department | CE, Industry 4.0 |

We received access to all Feralpi Group sustainability reports, balance sheets, and environmental certifications both at the Italian and German sites, in order to triangulate the answers given by the experts [97]. Based on the literature review, we developed a semi-structured questionnaire with three central parts: the first question obtains exploratory personal feedback on the definition and meaning of sustainability; the central part is an in-depth insight into the sustainability strategy of the organization, enabled by Industry 4.0 technologies and with practical implications on the operative business model; the last question is related to the possible future plan of action concerning sustainable and digital development, also considering the value chain. For business partners of Feralpi (including the customer and suppliers), the questionnaire was enriched with two additional points, in order to have a perception of the sustainability and Industry 4.0 strategy on the collaborative dimension in the value chain, which can also impact a company business model [101]. Business partners' perspectives helped to complete the view on the topic and reach data saturation. In Appendix A, we provide the two questionnaires more in detail. Interviews were transcribed and openly coded, with more detailed concepts on the first level and then with more synthetized themes [102]. Interviews were conducted in the interviewees' mother tongues (Italian and German). Professional certified translations of the interviews from Italian and German into English were carried out. In the following section, we present the main findings.

## 4. Main Findings

### 4.1. Environmental Sustainability as the Key to Successful Business Performance

Interpreting the challenges and benefits of Industry 4.0 related to a sustainability strategy means first getting insight into the perception of the concept of sustainability in the steel sector. The majority of respondents agree on the definition of sustainability as durable industrial activity, implying inclusive growth, encompassing all the triple-bottom-line dimensions (economic, environmental, social). Sustainability is a change of paradigm, both for the management and supply-chain partners ("The definition of sustainability on three dimensions -economical, environmental, social-is overcome: sustainability is now a company 360-degree strategy.", Feralpi Group, Interviewee K; "Sustainability is a change of paradigm: it means an inclusive, participated, equal growth, a durable economic business activity, which creates value and does not only optimize resources.", Feralpi Group, Interviewee I).

However, sustainability in the steel sector has its central core in the Circular Economy [62] and eco-friendly strategies, enabling the operative business of this sector ("Until the 90′s the economic dimension was predominant in our sector. Then important challenges were faced (such as plants dismantling due to overcapacity) and important restructuring actions were made concerning environmental sustainability.", Feralpi Group, Interviewee G).

Results underlined, in particular, how even more a clearer EU taxonomy could help companies in investing in environmentally sustainable activities, in order to calculate economic returns and to plan technical changes more precisely. The first delegated act on sustainable activities for climate change was published in April 2021; the second act was published on 9 March 2022, including energy and gas activities to accelerate the path

toward a carbon-neutral economy [103]. EU Taxonomy was partially criticized since the scope of action was too narrow and did not consider the carbon footprint of Small and Medium Enterprises (SME), the heart of the EU economy [62,104].

Sustainability has to be fully integrated into the company business model and create value for all stakeholders [18], which is confirmed both from the German Feralpi perspective and in the view of its customer ("Sustainability is a long-time company perspective, involving all three dimensions (economical, environmental, social).", Feralpi Germany, Interviewee R; "Sustainability is now completely incorporated in the company structure.", Customer, Interviewee B). In particular, looking at the perspective of production managers, sustainability "fulfills current generation needs, without compromising future generation exigencies" (Interviewee O), which is very close to the definition given in the famous Brundtland report, establishing the roots of the first definition of sustainability [15]. This is confirmed also by the perspective of environmental managers, underlying how the slogan of Feralpi's founder already contained the company's sustainability strategy since its origin: "Sustainability is contained in the motto of our founder: produce and growth with respect for human beings and the environment." (Interviewee M). These perspectives highlight how for operative business (production and environmental management) also, sustainability is a complete paradigm, acting simultaneously on all its dimensions [105].

For Feralpi managers and their business partners, the economic dimension of sustainability cannot exist without the environmental dimension and vice versa. CE practices are the environmental dimension of sustainability [106] ("First of all, sustainability means environmental sustainability, by employing all the best available technologies to be eco-friendly.", Feralpi Italy, Interviewee O; "Sustainability implies ecological sustainability, it means applying Circular Economy practices.", German supplier, Interviewee F). Circular Economy has become a great vehicle of business opportunity [107] since raw materials are becoming more and more expensive and scarce, the EU climate-neutral strategy is pushing the steel sector to find new paths of competitiveness [108] ("Raw materials have become more and more expensive, more costly. We have to better exploit renewable resources. Waste can get new life."—Interviewee N, Feralpi Italy).

As underlined by a digital service business partner of Feralpi, the social dimension of sustainability also plays a fundamental role in the economic strategy of a company in this industrial sector, by creating a workplace where employees can be personally and professionally satisfied ("We are responsible for our employees. We have to listen to the needs of our people. We have to know their ambitions, their expectations." (Interviewee D). Sustainable business models definitely create new economic value while facing environmental and social challenges [109,110].

The overall sustainability strategy of Feralpi has grown with time and encompasses both environmental and social dimensions ("We started our sustainability journey in 2004. At that time, we were even criticized, since it was as if we wanted to show off. Actually, environmental sustainability belongs to our DNA. We only want to be transparent, to show our culture, values.", Feralpi Group, Interviewee K). It is deeply rooted in its environmental commitment and it has constantly grown to permeate all stakeholders' dimensions. Today, Feralpi engages in seven specific pillars (SDGs) of the Agenda 2030, which are reconciled with its current industrial plan for the next five years and cover all three dimensions of sustainability:

SDG 6: Clean Water and sanitation.
SDG 7: Affordable and Clean Energy.
SDG 8: Decent Work and Economic Growth.
SDG 9: Industry, Innovation, and Infrastructure.
SDG 12: Responsible Consumption and Production.
SDG 13: Climate Action.
SDG 17: Partnerships for the Goals.

Feralpi actively participates in the United Nations Global Compact (UNGC), an internationalized network of 18.000 companies from 173 countries with the objective to foster the achievement of the 17 SDGs, and it is a founding member of the Global Compact Italian Network [111] ("Our 7 SDGs are oriented to a Circular Economy approach. We want to multiply the use of our materials, to have less waste and even more regenerated resources.", Feralpi Italy, Interviewee M).

Business partners of Feralpi confirm its sustainability strategy, especially concerning the environmental dimension ("I have observed Feralpi for long. I can see its full engagement in the ecological transition. I have complete esteem for its sustainability initiatives.", Italian customer, Interviewee B). The current path of Feralpi, as a family company, involves structuring itself in its governance like corporations listed on the stock exchange, with more formal internal procedures and sustainability independent experts, and allowing members to contribute their expertise to the Board of Directors ("We can take as governance model also companies outside our sector. Such companies settled an independent expert for digitization and another for sustainability in the Board, to give constant contribution in expertise fields.", Feralpi Group, Interviewee L). Corporate governance, intended as a structured system of processes, procedures, and responsibilities is a key element for sustainability, together with transparent non-financial disclosure [112].

The environmental strategy of Feralpi is fully intertwined with the economic perspective. Feralpi Group was the first steel company to have signed a positive loan of €20 million in 2019 to finance sustainability projects, and in 2021, it signed the first Circular-Economy-linked Interest Rate Swap loan worth €40 million, linked to the fulfillment of Circular Economy projects [113] ("This Circular loan has a more favorable interest rate than the first positive loan linked to ESG criteria. Among the KPIs that we monitor are the reduction of $CO_2$ emissions/produced tons, the increase of scrap suppliers selected through ESG criteria, the increase in percentage of recovered waste and strengthening of our compliance model to make our corporate governance more transparent", Feralpi Group, Interviewee L). The Feralpi Group is committed to a five-year sustainability Group strategy, where decarbonization, renewable energy, and CE play a central role. In 2021, company investments were 96% eligible for the EU taxonomy; indirect $CO_2$ emissions Scope 2 resulting from electricity were 10.35% less than those in 2019, with an annual goal of 90,000 t/a $CO_2$ reduction at the full operation of innovative technical investments (such as an induction electrical furnace for the rolling-mill plant both in Italy and in Germany instead of the gas consumption); there was an 8% increase compared to 2019 in recovered, recycled, and reused waste reuse on the total waste generated by steel processing [114].

*4.2. Industry 4.0 Concrete Applications: Reconciliation with Sustainability Strategy*

Concerning Industry 4.0 technologies, most interviewees confirmed that they are enablers of sustainable production, with a particular focus on the environmental strategy [115], although they are not exclusive to technologies supporting sustainable growth ("In sustainability projects, we can identify two types of enabling technologies: digital and Industry 4.0 technology, and process and technical innovation, which has not only to do with Industry 4.0.", Interviewee N, Feralpi Italy). This is an interesting point of view since environmental sustainability can be achieved through a mix of innovative technologies, enabled by digitization [116]

In particular, Industry-4.0-relevant technologies for the Feralpi Group are simulations, IoT, machine learning, and Big Data analysis, which can support environmental strategy in energy efficiency, monitoring of $CO_2$ emissions, traceability, optimization of resources, and employment of renewable materials in an optic of Circular Economy. The literature has already underlined the strict relationship between Industry 4.0, Circular Economy, and sustainability goals [117]. Thanks to the use of the IoT and Artificial Intelligence, the production process is more integrated and more monitored [118]. Industry 4.0 is an enabler of sustainability strategy, especially in the dimension of environmental protection and reduction of emissions [75]. The Cloud is an important basis for internal collaboration,

while for Feralpi Group business partners, it could acquire a pivotal role in integrating information in the supply chain and then enhancing the economic dimension by making the supplying process more efficient. The Cloud has also enhanced the social benefits of digitization [119], with the opportunity to utilize working from home, which was a powerful tool during the lockdown time due to COVID-19. Concerning the social dimension, autonomous and collaborative robots are important enablers of work safety for Feralpi Group suppliers and customers, while Augmented Reality could become a strategic opportunity to enhance safety [120], maintenance, and job rotation training, according to managers. In conclusion, cybersecurity has become an important pillar in the Feralpi strategy in terms of the implementation of Industry 4.0: secure interconnection can guarantee business continuity considering the economic and social sustainability dimensions. In this sense, we can speak in terms of cyber-resilience, creating an organizational and technical system of processes and procedures, which can maximize downtime reduction and minimize damages to operations in case of cyberattacks [121]. In Table 3, we present a synthesis of the main Industry 4.0 technologies chronologically adopted at Feralpi Group, focusing on their perceived sustainability benefits for managers, considering their impact on the three dimensions of the triple bottom line.

**Table 3.** Main Industry 4.0 technologies used at Feralpi Group and related sustainability benefits.

| Main Industry 4.0 Technologies Used at Feralpi Group and Related Sustainability Benefits | | |
| --- | --- | --- |
| **Type of Technology (Date of Adoption)** | **Triple Bottom Line Level** | **Main Sustainability Benefits** |
| Simulations | Economic and Environmental | More efficient production process, enhanced product quality, and optimization of resources |
| Advanced Robotics | Economic and Social | Safer and more comfortable workplace, enhanced productivity |
| Cloud | Economic and Social | Enhancing collaboration and facilitating information integration, exchange and transparency, among departments, subsidiaries, and potentially with suppliers and customers |
| IoT | Economic and Environmental | Monitoring of emissions and energy parameters |
| Machine Learning and Artificial Intelligence | Economic and Environmental | Production phases integration and quality enhancement, energy consumption, resource optimization |
| Big Data Analysis | Economic and Environmental | Energy consumption |
| Cybersecurity | Economic and Social | Guarantee of business continuity, intended in an economic perspective and in protection of the workplace from cyberattacks |

Trying to dig deeper into the different perspectives of the value chain, answers underline how suppliers are more focused on the internal application of IoT and robots, which have benefits for the standardization of processes and, in particular, in the case of robots, of increased comfortability at the workplace. Similarly, Big Data Analysis and Cloud and digital platforms not only enhance internal processes but also to build a bridge of communication among suppliers, producers, and customers.

Nevertheless, some interviewees (both internally in Feralpi Group and externally in the supply chain) highlighted the fact that sustainability benefits can be retrieved also with other types of technologies, which do not exactly belong to the cluster of Industry 4.0 or

digital innovation ("We have both Industry 4.0 technologies and other types of innovation, which can enable sustainability strategies.", Feralpi Group, Interviewee J; "We do not have real Industry 4.0 monitoring systems. We are going to first centralize our data with a structured ERP (Enterprise Resource Planning) system. We could look in the future to exchange data with our suppliers and customers.", Italian customer, regenerating black slag, Interviewee A, Feralpi Italy).

Looking at the implementation of innovative technology and sustainability projects, interpersonal networking, people awareness, and openness to change are fundamental ingredients [122]. In this sense, the social dimension of sustainability becomes an intrinsic aspect to enable an environmental strategy ("Technologies are important, but people make the difference. We have to make people network." Italian Business Partner, Polymers, Interviewee E).

The main opportunities of Industry 4.0 in sustainability strategy are bound to the economic dimension (improved productivity and efficiency, cost and resources optimization), which is confirmed by the current literature [123], but the creation of new partnerships in the value chain beyond the traditional consolidated business relationship is perceived as one of the greatest opportunities and it is an interesting novelty in our findings.

The characteristic of networking seems to unify the concept design of both Industry 4.0 and sustainability, especially concerning the use of innovative regenerated materials. Often these new networks are generated through interpersonal contacts, arisen by sharing similar sustainability values and strategies in their respective organizations ("We did not belong to the supply chain of Feralpi. We were in a completely different sector, that is plastics recycling. We knew Feralpi through its environmental managing director and then we started to talk about our collaboration.", Italian Business Partner, Polymers, Interviewee E).

Feralpi has been experimenting with the use of alternative resources instead of coke in the steel melting process, in order to reduce $CO_2$ emissions. The encounter with this supplier not belonging to the habitual supply chain was significant because the supplier had tried to diffuse the use of polymers instead of coke in some other steel producers, but there was a general response of skepticism, except for one established internationalized special steel producer. The experimental approach of the Feralpi management could enable the creation of a pilot technical process to enhance environmental sustainability by sacrificing the economic advantage at the beginning ("In Feralpi we found a young team, with a dynamic approach for experimentation. We were new to the process of the electric arc furnace. We started from scratch . . . At the beginning, the model was neither economical nor successful.", Italian Business Partner, Polymers, Interviewee E).

The basic ingredient of sustainable projects seems to be an open and experimental attitude, where technologies are also not considered certain drivers for success ("Sustainability projects do not at all give any guaranteed results at the beginning. They imply experimentation… Also some technologies are completely new and still not tested… You have to have a medium, long-term perspective, very different from the perspective of the linear economy.", Feralpi Italy, Interviewee N). Engwall et al. (2021) talk about an "experimental network", intended to be "a group of organizations collaborating in a time-limited, cross-industry network to explore potential business models for an anticipated, profound change in sociotechnical systems" [124] (p. 2). Actually, what could be observed in the case of the Feralpi Group is that those new experimental networks start to create a prototype, then they last over the period of experimentation and they continue, contributing to a more traditional framework of projects ("The polymer supplier involved us in another interesting project: thanks to DI.MA Interti (our participated company for the application of black slag in the concrete industry), we will deliver our Qubeco blocks for the new production site they are building. Moreover, we are talking about delivering together regenerated material to an important Italian market player for concrete mixture and paving.", Feralpi Italy, Interviewee M).

As strongly underlined by Da Giau et al. (2020), a sustainability strategy should be a gatekeeper for manufacturers to find opportunities and support the creation of strategic alliances to capture change in the business context [125].

As highlighted by Feralpi business partners, the sustainability strategy enabled by Industry 4.0 technologies can integrate a company into a broader network, reaching even its own competitors [126]. In this sense, the social dimension of sustainability is reached on the highest level of its fulfillment, as also represented in SDG 17—partnerships for the achievement of the goal. An interesting example of partnership, looking at our case study, is the creation of "Management 4 Steel" in 2019, an academy for the steel sector, which was initiated by four Italian regional steel producers. Four competitors decided to collaborate and network in the domain of advanced education ("We want to see what others are doing. All of us—although competitors—need competent young people and need to create and maintain competences in our sector.", Feralpi Group, Interviewee K).

In the paradigm of Industry 4.0 and sustainability, benefits and opportunities can, at the same time, be challenges. The management and employees represent a key turning point in the implementation of digital innovative technologies and sustainable business models [127]. Industry 4.0 can become a set of unrelated unsuccessful technological invest-ment decisions bringing confusion and inefficiency if it is not coherently integrated into the long-term company economic and environmental strategy. If sustainable business projects have an experimental character, they can end in useless and ineffective results if they are only driven by technological choices and not by concrete needs. People are the key to successful sustainable innovative patterns ("We need motivated, dynamic people… We want to become the Ferrari of the steel sector.", Feralpi Group, Interviewee K; "I would need more internal people with interdisciplinary competences in Industry 4.0 and sustainability, like you, rather than consultants who one day will go to other customers. We need a spark. Time. An occasion to sit together at a table.", Customer, Feralpi Group, Interviewee B).

For Feralpi managers and business partners, the main challenges consist not only of the technical implementation and integration of Industry 4.0 in the plant or in the intensive financial capital needed to turn prototyping projects into successful business cases, but also of the cultural and organizational shift accompanying the digital transformation and the time and human resources to be rightly allocated for the innovation process [54]. The need for new multidisciplinary, hybrid technical and soft-skill competencies is confirmed by the existing research [53], as both an opportunity and challenge in terms of the fulfillment of sustainability strategy in digital transformation contexts [128] ("Sustainability is a great opportunity to make our management grow, to uplift their competences and capabilities.", Feralpi Group, Interviewee K). Industry 4.0, together with sustainability, requires a new managerial leadership profile, where decisions are decentralized in small teams, rather than a central authority [129].

Sustainability and Industry 4.0 practices enhance another aspect of maturity for the management: the introduction of more structured governance in order to monitor and control procedures and investments decisions [130] ("Sustainable strategies imply that we structure ourselves with a more defined, transparent governance, like other corporations.", Feralpi Group, Interviewee L).

### 4.3. Sustainable Business Models through Supply Chain Integration: The Role of Industry 4.0

Industry 4.0 and sustainability practices are modifying the business model of the steel industry. According to German Feralpi managers, some disruptive business patterns will change the commercial business model of steel producers, especially for commodities manufacturers like Feralpi Group. In this case, with the use of digital platforms and e-commerce, the central figure of traders will be doomed to vanish with time since final customers will be able to directly reach original producers, enhancing the control of the quality of material and taking advantage of a shorter value chain ("The e-commerce will come also for our sector, as already happened in the automotive industry and in the B2C—Business To Consumer sector. It is not something new. It was already introduced in the

past ("Steel 24/7"), but it didn't work. One day, when it will work for us, too, it will be disruptive." Feralpi Germany, Interviewee R). Industry 4.0 can maximize a company's financial performance, by improving both the internal business process performance (IBPP) and the whole supply chain performance [131].

In Figure 1, we summarize the business relationships in the steel commodity sector.

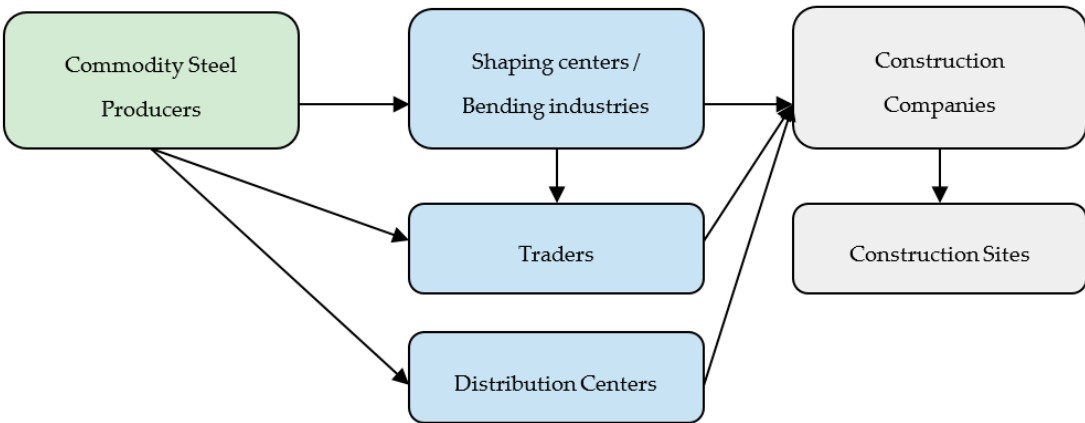

**Figure 1.** Commercial distribution cycle for commodity steel producers.

Concerning the Business Unit of Specialties of Feralpi Group, digital platforms will foster data exchange between suppliers and customers, with improved transparency and product quality. Actually, future projects, which will eventually exploit NGEU funds, arenot only focused on the implementation of digital and innovative technologies in the organization but also on becoming a facilitator integrating and consolidating information into their supply chain and beyond [76]. However, the lack of reciprocal trust in the use of data and the lack of maturity in cybersecurity protection systems in the supply chain hinder the initialization of such projects ("Data exchange in the supply-chain is important both in the sector of commodities and special steels. As producers, we could give an additional service to our customers, compared to our competitors. However, there is often a speculative environment. Transparency could also mean being exploited by your business partner. Moreover, data are sensible for us. Cybersecurity is still an open topic.", Feralpi Group, Interviewee G).

Industry 4.0 and sustainability strategy are opening steel industries to new possible business model paradigms, where producers can also come closer to the end customers and where byproducts can become an employable resource both in their own production cycle and in other sectors [132]. However, some interviewees' opinions underline how Industry 4.0 and sustainability are improving the profitability, performance, and environmental impact of steel manufacturers. However, radical business models transformation is not really being observed since the steel production process and market constraints are external factors to be coped with in the best way possible ("Processes will be more and more efficient. Yield will improve. We will optimize our emissions impact more and more. But steel will always be melted at 1.560 degrees.", Feralpi Germany, Interviewee Q; "We improve our costs, our profitability, we can reach new markets, but basically our business model remains the same.", Feralpi Italy, Interviewee N; "We will further make steel.", Feralpi Italy, Interviewee O).

### 4.4. Final Discussion: Industry 4.0 Benefits and Sustainability Strategy

This research contributes to the current literature studying the impact of Industry 4.0, as related opportunities and challenges in the context of sustainability strategy for manufacturing industries, especially considering the realm of energy-intensive producers, and how some of these technologies are related to the economic, environmental, and social domain of sustainability. The novelty of our study is to investigate these aspects,

especially, from a managerial point of view in a sector fundamental to the EU economy, i.e., steel manufacturers. Steel industries could be dramatically affected by the sustainability paradigm pushed by the EU if they have yet to start investing in innovative technologies and business partner networks, in order to fulfill demanding environmental goals. We saw how different Industry 4.0 technologies (i.e., simulations, robotics, IoT, Cloud, Big Data Analysis) can enhance sustainability strategies on different dimensions and with different benefits:

- Environmental: more efficient production process and use of resources, enhancing CE, energy efficiency, and $CO_2$ emissions reduction practices;
- Social: safer working conditions and higher data transparency and exchange in and outside the organization;
- Economic: enhanced efficacy in technical and organizational processes, including the employment of more affordable regenerated raw materials.

Industry 4.0 can become a successful enabler of a long-term sustainability strategy, which is intertwined with the development of Circular Economy patterns [133,134]. The transformation of costly disposal materials into valuable byproducts is a paradigm of how a new business model can be created thanks to the contribution of new business partners of different sectors and expertise. This can build the basis for open innovation contexts, where managers of manufacturing industries do not fear the collaboration of unexpected stakeholders, even competitors [135], and take sustainability and Industry 4.0 contexts as an opportunity to develop new managerial capabilities.

In this sense, company capabilities in the context of sustainability, with particular attention to the environmental dimension, become a key competitive advantage [136]. It lies in the dynamic capabilities of organizations to seize the opportunity of technologically sustainable projects. It is fundamental for energy-intensive industries to create internal routines that uniform the capabilities of managers to guide sustainability strategy and Industry 4.0 projects successfully, to sense and seize market opportunities, and to manage the correlated risks [137].

This paper contributes also to the current literature on sustainable business models, which is a broad field for incremental and disruptive innovation since the organizational and technological path to sustainability for energy-intensive industries is still an experimentation field. Sustainability can bring companies new opportunities for value creation (new values for existing stakeholders or new stakeholders), but if companies do not focus on the right strategy, the existing value can also be destroyed [138].

With Industry 4.0, steel producers come closer to their customers, and their suppliers want to support them in their digital transformation in the context of sustainable growth. We also saw how, in the case of commodity producers, Industry 4.0 can enhance customer centrality. The new value proposition of sustainability to have inclusive growth in all three dimensions (economical, environmental, social) is also becoming important for customers in energy-intensive sectors because current norms are pushing them to find alternative material sources in their production model.

Thanks to the analysis of a company case in a special context put under pressure by strong competition forces and high environmental constraints, this paper can offer managers some insights into a successful model for sustainability strategy with the following key aspects: conscientious use of some Industry 4.0 technologies, openness to new partner networks in the supply chain, trial and error approach in experimental projects, and enhancing interdisciplinary competences of employees.

The experimental character of sustainable paths is a fundamental element to develop a successful approach of management to business transformation and business model innovation [139]. However, this represents a consistent challenge for the management since this implies investments with new technologies, intensive financial capital, and no guarantee of success, especially in the short-term perspective.

This gives further concrete insight into the complex process of business model innovation and its key success factors and its challenges [140], which are mostly related to the competencies of the own management and employees.

## 5. Conclusions

The sustainability paradigm pushed by the EU should not fear European manufacturers, including those that are in an energy-intensive sector, such as steel industries.

Industry 4.0 can give energy-intensive manufacturers the opportunity to embrace long-term economic, environmental, and social sustainable growth, strengthening the competitive advantage in the current market, by creating diverse business partnerships and collaborations to reach new markets. Further research should explore the nature of these collaborations in a deeper way and look for some elements acting as facilitators, such as geographical nearness or shared specific knowledge.

Sustainability and digital innovation would not only lead to consolidation of the competitive advantage of those industries but they could change their business model radically in a positive way if they are open to embracing organizational and technological change, facing entrepreneurial courageous decisions, and finding new paradigms for technological experimental applications. In this context, there is an open domain of research, since it is still difficult to capture the evident effects of technological impact on the sustainability strategy of manufacturing industries [133]. It is still being debated how companies can build an assessment model with a set of precise indicators [134] in order to capture the benefits of sustainability strategy on the three dimensions (economical, environmental, social) for their business model. Economic KPIs (Key Performance Indicators) are often used to measure sustainability performance, although they are not sufficient. Our research has the limitation of not deepening the analysis with quantitative data and indicators associated with Industry 4.0 technology application in a sustainability strategy. Further research should focus on the measurability of such advantages and the exploration of success factors of sustainable business models and supply chain integration. We acknowledge that our results are at the beginning of understanding the paradigms of Industry 4.0 and sustainability for energy-intensive manufacturers, but we believe that a further deepening analysis of such industries, not only in the environmental but also in the social-economic dimension of the triple bottom line, could provide the impulse for other sectors to build a successful path of digital and sustainable growth.

**Author Contributions:** All authors have contributed to all phases of the manuscript. All authors have read and agreed to the published version of the manuscript.

**Funding:** This research received no external funding.

**Data Availability Statement:** Not applicable.

**Conflicts of Interest:** The authors declare no conflict of interest.

## Appendix A

APPENDIX 1: Questionnaire for Feralpi manager
APPENDIX 2: Questionnaire for business partners
APPENDIX 1: QUESTIONNAIRE TO ITALIAN AND GERMAN FERALPI MANAGERS

(1) What does the term "sustainability" mean for you and which role does it play in your company?
(2) What are, in your opinion, the essential objectives of your company, considered the economic, environmental and social sustainability dimension? Which is the most relevant dimension?
(3) Which Industry 4.0 technology does enable sustainability objectives? What are the challenges and opportunities? Are opportunities measurable?

As a reminder, Industry 4.0 technologies are: —collaborative and autonomous robotics; simulations; vertical integration (e.g., in the production, advanced integrated automation, machine learning, artificial intelligence); horizontal integration (e.g., in the supply chain, blockchain protocols and/or digital platforms to share data); IoT; cybersecurity; cloud; additive manufacturing; virtual and augmented reality; big data analysis.

(4)　How has the sustainability strategy modified or will modify the business model of your company?

(5)　How will the sustainability and Industry 4.0 strategy develop in your company in the next future? Do you already have related strategic projects and do you think you should apply for NGEU funds (as a reminder Next Generation EU funds represent almost €2 billion for EU members in order to recover from the COVID-19 pandemic crisis)?

APPENDIX 2: QUESTIONNAIRE TO ITALIAN AND GERMAN FERALPI BUSINESS PARTNERS

(1)　What does the term "sustainability" mean for you and which role does it play in your company?

(2)　What are, in your opinion, the essential objectives of your company, considered the economic, environmental and social sustainability dimension? Which is the most relevant dimension?

(3)　Which Industry 4.0 technology does enable sustainability objectives? What are the challenges and opportunities? Are opportunities measurable?

As a reminder, Industry 4.0 technologies are:—collaborative and autonomous robotics; simulations; vertical integration (e.g., in the production, advanced integrated automation, machine learning, artificial intelligence); horizontal integration (e.g., in the supply chain, blockchain protocols and/or digital platforms to share data); IoT; cybersecurity; cloud; additive manufacturing; virtual and augmented reality; big data analysis.

(4)　How has the sustainability strategy modified or will modify the business model of your company?

(5)　How will the sustainability and Industry 4.0 strategy develop in your company in the next future?

(6)　Do you already have related strategic projects and do you think you should apply for NGEU funds (as a reminder Next Generation EU funds represent almost 2€ billion for EU members in order to recover from the COVID-19 pandemic crisis)?

(7)　Which projects realized with Feralpi could enable the sustainability objectives of your company? Could be they achieved thanks to Industry 4.0 technologies?

(8)　Which kind of projects have you in plan with Feralpi for the future concerning sustainable and digital development? Do you think you should apply for NGEU funds?

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
