# Peer review of "The Impact of Industry 4.0 on the Steel Sector: Paving the Way for a Disruptive Digital and Ecological Transformation"

_recycling, doi:10.3390/recycling8040055_

Round 1

Reviewer 1 Report

The author should mention the novelty of the work compared to other works.

Table 1 should be put together in one sheet

The abstract should be improved because it is ambiguous.

- The bibliography should be updated because there is little updated bibliography.

- The figures and tables should be improved, and should be in an adequate and legible format.

Regarding the objective of the work, it should be highlighted, as well as the working hypothesis. On the other hand, emphasis should be given to the innovation of the work and its specific contribution, and the conclusions should be improved.

On the other hand, in Figure 5. Life Cycle of Steel (Kawai, 2001) mentions deposits and mining, but does not explain how it would indicate the impact of Industry 4.0 with them in its ecological transformation.

Author Response

Dear reviewer, thank you very much for your suggestions.

Here we have or responses:

a) Novelty: we better highlighted the novelty of our work in the introduction and literature review

b) Table 1 was put together in one sheet

c) Bibliography was updated

d) Figures and tables were improved in their quality

e) we specified the contribution and structured conclusions in a better way

f) we wliminated the fig. of kawai. we describede the concept better in the text.

Regards

Reviewer 2 Report

The Impact of Industry 4.0 on the Steel Sector: Paving the Way for a Disruptive Digital and Ecological Transformation

Article is interesting and well written. Few comments are suggested to further improve the quality of the article.

Abstract must be revised to highlight the imporance of this study.

Introduction must be further updated with the latest sutdies.

Under green economy, green concrete must be inlcuded in the introduction by included following studies in the reference.

Khan, K., Amin, M. N., Saleem, M. U., Qureshi, H. J., Al-Faiad, M. A., & Qadir, M. G. (2019). Effect of fineness of basaltic volcanic ash on pozzolanic reactivity, ASR expansion and drying shrinkage of blended cement mortars. Materials12(16), 2603.

Khan, K., Ahmad, W., Amin, M. N., & Nazar, S. (2022). Nano-silica-modified concrete: A bibliographic analysis and comprehensive review of material properties. Nanomaterials12(12), 1989.

Please provided references of figure 2 if taken from other sources.

Table at page 5 must had table number, caption and citation in text.

The quality of figures 4 and 5 is poor.

Discussions must be referred with the previous studies publised in the existing studies.

Conlcusions must be concise and to the point.

Author Response

Dear reviewer, thank you very much for your suggestions.

a) Abstract was revised and made more concrete

b) Introduction was updated with the latest studies

c) we added your suggestions in the bibliography under the topic green concrete

d) we improved and better cited figures and tables

e) we improved the cited literature in main findings and discussions

f) we improved the conclusion with limits and further research, making it more concise.

Regards

Reviewer 3 Report

Page 18, last sentence in 4.3 is not finished "...Industry 4.0 maximizes the company's financial performance, by affecting both the internal business process performance (IBPP) and"

Please take a look at tables and texts inside them an reformat it in a way that the words will not be split between lines.

It would be valuable for the reader if some financial or man power data would be provided (in %) as a cost of changes or expected future profits of company approach to new challenges.

It would be also important to define some quantitative factors for company transformation and use for example surveys for gathering data and more statistic approach to the problem.

I know it is not possible for now, but it would be great if in future research comparison between at least two companies working in the same industry is performed.

Author Response

Dear reviewer, thank you for your suggestions

a) In chapter 4.3 we completed the sentence

b) we corrected text in tables so that it is more readeable and organized

c) we improved some absolute and % numbers about the significant investments of the analyzed company

d) the deployment of more quantitative factors and indicators is planned to be done in a further research phase, as prolongation of the qualitative study

e) we will take your suggestion of comparison with another industrial similar case in our future research.

Regards

Reviewer 4 Report

1 Please clearly illustrate the innovation of current research. It lacks innovation.

2 The research should be more than an interview.

Author Response

Dear reviewer, thanks for your suggestions

a) We better highlighted the innovation of our study in the introduction and in the literature review

b) there was a misunderstanding: interviews have always been 19. In the transcription of the citations, to make them more anonimous, we put only manager without the area. We corrected this.

Regards

Round 2

Reviewer 4 Report

The manuscript is still an interview rather than a paper.

Author Response

Dear Reviewer, than you for your additional suggestions.

We read again your first and second review, and we reread our paper, improving all chapters for give a sense of more clarity. Especially in findings, we rearranged the presentation, so to give more a compact view on results on the 19 interviewes we conducted.

Our text is currently under English proof reading.

regards